# Electronic Structures of Radical-Pair-Forming Cofactors in a Heliobacterial Reaction Center

**DOI:** 10.3390/molecules29051021

**Published:** 2024-02-27

**Authors:** Yunmi Kim, A. Alia, Patrick Kurle-Tucholski, Christian Wiebeler, Jörg Matysik

**Affiliations:** 1Institut für Analytische Chemie, Universität Leipzig, Linnéstraße 3, D-04103 Leipzig, Germany; yunmi.kim@uni-leipzig.de (Y.K.); patrick.kurle@uni-leipzig.de (P.K.-T.); christian.wiebeler@uni-a.de (C.W.); 2Leiden Institute of Chemistry, Leiden University, Einsteinweg 55, 2301 RA Leiden, The Netherlands; alia.aliamatysik@medizin.uni-leipzig.de; 3Institut für Medizinische Physik und Biophysik, Universität Leipzig, Härtelstr. 16-18, D-04107 Leipzig, Germany; 4Institut für Physik, Universität Augsburg, Universitätsstraße 1, D-86159 Augsburg, Germany

**Keywords:** solid-state photo-CIDNP, solid-state NMR, HbRC, electronic structure

## Abstract

Photosynthetic reaction centers (RCs) are membrane proteins converting photonic excitations into electric gradients. The heliobacterial RCs (HbRCs) are assumed to be the precursors of all known RCs, making them a compelling subject for investigating structural and functional relationships. A comprehensive picture of the electronic structure of the HbRCs is still missing. In this work, the combination of selective isotope labelling of ^13^C and ^15^N nuclei and the utilization of photo-CIDNP MAS NMR (photochemically induced dynamic nuclear polarization magic-angle spinning nuclear magnetic resonance) allows for highly enhanced signals from the radical-pair-forming cofactors. The remarkable magnetic-field dependence of the solid-state photo-CIDNP effect allows for observation of positive signals of the electron donor cofactor at 4.7 T, which is interpreted in terms of a dominant contribution of the differential relaxation (DR) mechanism. Conversely, at 9.4 T, the emissive signals mainly originate from the electron acceptor, due to the strong activation of the three-spin mixing (TSM) mechanism. Consequently, we have utilized two-dimensional homonuclear photo-CIDNP MAS NMR at both 4.7 T and 9.4 T. These findings from experimental investigations are corroborated by calculations based on density functional theory (DFT). This allows us to present a comprehensive investigation of the electronic structure of the cofactors involved in electron transfer (ET).

## 1. Introduction

Using solar energy, photosynthesis provides the basis for a sustainable life of human species by providing oxygen and food. Solar conversion efficiency of natural photosynthetic organisms is very high, and the quantum yield for the light-induced electron transfer (ET) in natural reaction centers (RCs) is almost 100% [1,2]. Consequently, research into meeting our considerable future energy needs is aimed at gaining a comprehensive understanding of this process.

There exist two categories of natural RCs, distinguished by their terminal electron acceptors. Type-I RCs occur in anaerobic heliobacteria, green sulfur bacteria, as well as oxygenic photosystem I in cyanobacteria and plants utilizing iron–sulfur ([4Fe-4S]) clusters, whereas type-II RCs exist in purple bacteria and photosystem II (PSII) carrying quinone molecules as the terminal acceptor. Concerning the symmetry of the polypeptides, RCs can also be categorized as either homodimers or heterodimers [3]. While PSI and all type-II RCs are heterodimers, RCs of heliobacteria and green sulfur bacteria are homodimeric. 

The first family of heliobacteria, *Heliobacterium* (*H.*) *chlorum*, discovered in 1983 by Gest and Favinger [4], revealed a new form of pigment, bacteriochlorophyll *g* (BChl *g*). This cofactor exhibits a unique absorption spectrum and is sensitive to oxygen, reflecting an early stage in the evolution of chlorophyll [5,6,7]. Subsequently, *Heliobacillus* (*Hb.*) *mobilis* [8] and *Heliobacterium* (*H*.) *modesticaldum* [9] were isolated. Studies on protein structure, on biochemical and spectroscopic properties [7,10,11,12,13,14,15,16], as well as on the kinetics of the ET [16,17,18,19,20] were undertaken.

The recent report on the X-ray structure of the heliobacterial RC (HbRC) from *H. modesticaldum*, with a high resolution of 2.2 Å [21], settled questions remaining from earlier work (Figure 1 based on that X-ray structure). In the HbRCs, the cofactors involved in ET are arranged into two identical branches with *C*_2_ symmetry; therefore, it is proposed that the ET occurs equally along these two branches [15,21,22]. This phenomenon distinguishes them from other type-I RCs in cyanobacteria and photosystem I (PSI), where the A-branch is favored over the B-branch for ET due to the heterogeneity of the protein [3]. HbRCs carry three different types of special pigments: The primary electron donor, P800, is a dimeric supermolecule formed by two bacteriochlorophyll *g* epimer (BChl *g*′) cofactors [11], and the primary electron acceptor A_0_ is a chlorophyll (Chl) *a* type pigment, 8^1^-hydroxy-chlorophyll *a* with a farnesyl side chain (8^1^-OH-Chl *a_F_*) [12,13]. In between the donor and acceptor cofactors, the accessory cofactors BChl *g* are localized. Furthermore, as the terminal acceptor, the 4Fe-4S cluster Fx [14] is formed by conserved Cys residues [21,23]. Although menaquinone has not been observed in the X-ray structure, several working groups have reported the existence of a quinone as a secondary acceptor A_1_ [10,24,25], which is loosely bound between the primary acceptor A_0_ and the terminal acceptor Fx [21,26]. However, the P800^+^A_1_^−^ state during forward ET has not been definitively identified [27,28,29], and ET occurs in quinone-depleted membranes [30,31]. Notably, the P800^+^A_1_^−^ state was identified only under pre-reduced conditions of Fx and at temperatures of 5–20 K [32,33,34]. 

This perfect symmetry in polypeptide and cofactor arrangement makes HbRC both structurally and functionally homodimeric. The symmetric electronic structure of the oxidized special pair in HbRC was also obtained with density functional theory (DFT) calculations indicating ET through both branches of cofactors would happen with equal efficiency, as discussed in a recent review [3]. Because of this simplicity, it was hypothesized that the HbRC represents an evolutionarily primordial RC [21,35].

Among the various types of RCs, the diversity of the protein matrix and molecular structure and arrangement of cofactors results in notable variations in ET functionality as expressed, for example, by the redox potential [36]. P680 of PSII, having a redox potential of ~1.2 V, is known to be the strongest oxidizing agent in living nature [37,38]. On the other hand, P800 of HbRCs, having a redox potential of ~225 mV, is one of the strongest reducing agents in living nature [3,23,39]. It has been proposed that ancestral homodimeric RCs are adapted to their respective light–environmental conditions [3].

Upon light absorption by the light-harvesting complex, and excitation transfer to RC, the electronically exited primary donor reduces the primary acceptor A_0_, forming a spin-correlated radical pair (SCRP). Under natural conditions, the electron is transferred further to the terminal acceptor Fx. The pre-reduction of the terminal acceptor blocks the forward ET beyond A_0_; therefore, an SCRP is formed between the primary donor P800 and the primary acceptor A_0_ [40] (Figure 2). Depending on the spin-state of the SCRP, it either recombines to the electronic ground state or forms a donor triplet state. This donor triplet state has a lifetime of 35 µs at room temperature and even longer (100–400 µs) at low temperature [14,23,41,42,43].

The solid-state photochemically induced dynamic nuclear polarization (photo-CIDNP) effect [44] arises from the electron–electron–nuclear dynamics during the SCRP and the donor triplet state [45,46], which induces non-Boltzmann nuclear spin polarization. This effect can be detected in solid-state NMR experiments, resulting in highly enhanced signals originating from the electron donor and acceptor cofactors with an enhancement factor of 10,000 in ^13^C nuclei [47]. This appears to be a ubiquitous property of ET in all natural photosynthetic RCs [48]. The effect has been observed in various RC types, such as the type-II RCs of purple bacteria of *Rhodobacter* (*R.*) *sphaeroides* wild type (WT) [49,50] and its carotenoid-less mutant R26 [51,52], as well as in PSI and II from plants [53,54] and diatoms [55]. Also, homodimeric type-I RCs of *Hb. mobilis* have been studied [56,57,58]. Furthermore, this effect has been more recently observed in flavoproteins called LOV domains [59,60,61,62,63]. 

As in all photosynthetic RCs, under strongly pre-reduced conditions, the SCRP [P^•+^A_0_^•−^] in HbRC is created in its electronic singlet state (S) and undergoes free spin evolution into the electronic triplet (T_0_) state. Under solid-state conditions, this evolution is controlled by the secular part of hyperfine interaction (*A_zz_*), the difference in electron Zeeman frequency (ΔΩ), and the electron–electron coupling (*d*). During spin evolution, electron spin order is transferred to the nuclei through the contribution of the three-spin mixing (TSM) mechanism by matching of three energy levels requiring a triple-matching condition 2|ΔΩ| = 2|ω_I_| = |A| [64]. The mechanism is driven by the anisotropic pseudo-secular component of the hyperfine interaction (*B*), along with electron–electron coupling (*d*). The difference in the decay rate of the SCPR in its singlet and triplet states leads to nuclear polarization buildup in ground state, a mechanism known as differential decay (DD). In this mechanism, the matching condition is 2|ω_I_| = |A| [65,66]. As a result, the TSM and the DD mechanism serve as coherent mechanisms for transferring polarization from electrons to nuclei. In the present case, the difference in kinetics of the two decay channels is minor, and action of the DD can be neglected. If the donor triplet state lives sufficiently long, the nuclear hyperpolarization on the triplet decay pathway is relaxed by electronic interactions, while the hyperpolarization of opposite sign on the singlet decay pathway survives. This modification of the classical radical-pair mechanisms has been coined differential relaxation (DR) mechanism [48,65]. This mechanism exclusively affects only the signals from the donor cofactors. Hence, the photo-CIDNP intensities acquired during continuous illumination under solid-state conditions encompass three mechanisms (TSM, DD, and DR), making it challenging to separate and discern their individual contributions. Theoretical analysis on RCs of *R. sphaeroides* WT provides an explanation for the predominantly emissive (negative) spectral envelope, which arises from the prevailing influence of TSM over DD. In its carotenoid-less mutant R26 [51], enhanced absorptive (positive) signals, which are specifically observed from the donor, are explained by the DR mechanism.

By measuring the kinetics of the radical-pair decay of HbRCs by time-resolved fluorescence [67], ultrafast optical experiments [40], as well as laser flash photolysis [41], it becomes apparent that there is no substantial discrepancy in singlet and triplet decay channels of the SCRP, implying a very limited contribution of the DD mechanism [65,66]. The X-ray structure of HbRCs (PDB:5v8k [21]) exhibits two carotenoids within the protein matrix. Notably, it reveals a significant distance from P800 to the carotenoid molecule, measuring about 16.7 Å (center to center) and 14.1 Å (edge to edge). This differs from the situation in *R. sphaeroides* WT (PDB:1M3X [68]), where a shorter distance is observed at 14.4 Å (center to center) and 8.6 Å (edge to edge). Thus, the HbRCs lack an efficient triplet quencher such as carotenoids near the donor special pair [21], which allows for fast relaxation of the electronic donor triplet state [41,42]. Consequently, the lifetime of the triplet donor is significantly longer (35 μs) [14,23,41,42] compared to that in the RC of *R. sphaeroides* WT (100 ns) [50]. Hence, the positive signals assigned to the DR mechanism arise exclusively from the electron donor. These signals were previously observed exclusively at 4.7 T in a study on *Hb. mobilis* [58], which exhibits a similar donor triplet lifetime to the carotenoid-less mutant *R*. *sphaeroides* R26 [51]. Thus, in HbRCs, positive signals are due to the DR mechanism and assigned to the donor, while negative signals can be linked to the TSM mechanism.

Previous photo-CIDNP research on HbRCs primarily focused on comparing anaerobically treated *Braunstoff* and aerobically treated *Grünstoff* of *Hb. mobilis* in a broader magnetic field range [57,58]. A strong increase in the positive signals in *Braunstoff* towards lower fields was already observed, demonstrating a consistent ratio of positive signal intensities upon changes in the magnetic field. That work, as we recognize now, illustrates the strong field effect of the DR mechanism, as it is also has been observed in RCs of *R. sphaeroides* R26 [69]. Interestingly, in the *Grünstoff*, the DR appears to be quenched [57]. One might speculate whether molecular oxygen acts here as a triplet quencher. Despite these observations, ET occurs in these two operational states, suggesting the evolution of aerobic photosynthesis in Earth’s history.

The significant enhancement of NMR signals induced by the solid-state photo-CIDNP effect facilitates the analysis of the electronic structure of active cofactors. In HbRC under pre-reduced conditions, these active cofactors in the photo-CIDNP MAS NMR experiment are the electron primary donor, BChl *g*′, and the primary acceptor, 8^1^-OH-Chl *a_F_*_._ The NMR chemical shifts are indicative of the electronic structure of the ground state after the photocycle, while light-induced photo-CIDNP intensities are correlated with the local electron spin densities in the *p_z_* orbital during the lifetime of the SCRP [48,70]. However, a comprehensive picture of the electronic structure is still missing due to restrictions on sample preparation limiting signal intensity [56,57,58] and being confined to a magnetic field of 4.7 T [58], which, as demonstrated here, is not optimal for observing signals from the electron primary acceptor.

Hence, in the present study, we prepared three different types of selectively ^13^C isotope-enriched and uniformly ^15^N isotope-enriched membrane fragments of anaerobic *Braunstoff* of *Hb. mobilis*. The biosynthetic pathway for the selective ^13^C isotope labelling of chlorophyll cofactors using *δ*-Aminolevulinic acid (ALA) as the precursor of chlorophyll formation [71,72] is described in Figure 3. The aim of these endeavors is to re-construct the electronic structure of the active cofactors forming the SCRP by combining photo-CIDNP MAS NMR experiments for signal enhancement with calculations based on DFT. The elucidation of the electronic structure in the electronic ground state is based on chemical shift assignment, achieved via both 1D and 2D ^13^C-^13^C dipolar-assisted rotational resonance (DARR) photo-CIDNP MAS NMR experiments. These experiments are conducted at magnetic fields of 4.7 T and 9.4 T, respectively. By using different enhancement mechanisms, these experiments are tailored to optimize signal observation for the primary donor and acceptor cofactors, respectively. Additionally, we further investigate local electron spin densities of cofactors in the radical-pair state, as derived from photo-CIDNP MAS NMR intensities. The theoretical analysis is reinforced by Mulliken spin population analysis for the neutral donor triplet state and hyperfine anisotropy for the radical-pair anion state from DFT calculations.

## 2. Results

### 2.1. Isolation of Membrane Fragments

For optimizing the signal strength, the isolation of the photosynthetic membranes from cells is desirable. To explore whether the isolation induces a modification on the cofactors of the RC, we studied the effect of isolation on the electronic structure. The ^13^C MAS NMR spectra obtained from natural abundance of the intact cells and membrane fragments in both dark and continuous illumination at 4.7 T are represented in Appendix A. By comparison of the light-induced spectra collected from these two distinct sample preparations (Figure 4(Ba,Bb)), a notable increase in signal intensity is observed in the membrane fragments, which exhibits 26 light-induced signals. We noted a slight alteration at methine carbons at 97.1 ppm and 96.2 ppm, revealing a clear signal enhancement in the membrane fragments, which was assigned to C5 of the electron acceptor according to previous study [58]. This change might be attributed to a slight modification in the protein environment upon isolation. In addition to the remarkable chemical shift similarities, minor differences became apparent at 187.7 ppm and 127.4 ppm in membrane fragments, which were absent in intact cells. Nevertheless, the ratio of total observed absorptive peak areas to emissive peak areas between these sample preparations remains constant (the ratio for intact cells: 1.43, membrane fragments: 1.41), suggesting that the contribution of the enhancement mechanisms producing the solid-state photo-CIDNP effect remains unchanged.

Our data demonstrate that, upon isolation of membrane fragments from intact cells, the electronic structure as well as the radical-pair dynamics are highly conserved. This implies that the general cellular environment does not influence the electronic ground state of the central cofactors. 

### 2.2. Magnetic-Field Dependency 

Photo-CIDNP MAS NMR spectra at ^13^C natural abundance of membrane fragments obtained at two different magnetic fields, 9.4 T and 4.7 T, in the dark (Figure 4(Aa′,Ab′)) and under the illumination (Figure 4(Aa,Ab)) are shown in Figure 4. As expected, the solid-state photo-CIDNP effect in HbRCs is strongly dependent on the magnetic field. The difference of the light-induced signals obtained from these two magnetic fields is the occurrence of different phases implying two different mechanisms. Obviously, both the absorptive and emissive signals show increased signal intensity at 4.7 T and the absorptive signal occurs strongly at 4.7 T compared to the field at 9.4 T. That implies that the contribution of the DR mechanism arising from the donor triplet is increased at 4.7 T. It is a similar trend as it has been observed in previous study of magnetic-field dependency of RCs of *R. sphaeroides* R26, where maximum signal enhancement at lower magnetic fields in the range of 1.4–2.4 T has been observed [69]. Also, excluding the contribution of the DD mechanism, emissive signals can be attributed entirely to the TSM. 

The chemical shifts of the observed signals remain constant (Figure 4(Ba,Bb)), and the intensity ratio among the emissive signals shows slight change between two magnetic fields. The signal with the highest intensity at 134.2 ppm remains unchanged. In addition, the methine carbon signals at 108.7, 97.1, 96.2, 93.1, and 91.9 ppm exhibit more significant enhancements at 4.7 T, whereas these signals are not present at 9.4 T. This could be attributed to the low electron spin density observed on these methine carbons (i.e., C5, C15, and C20) in the donor triplet state, which was confirmed by the analysis of the Mulliken populations from the corresponding DFT calculation (see Appendix A), weakening the DR contribution. 

### 2.3. ^15^N Chemical Shift Assignment

The ^15^N MAS NMR spectra of uniformly ^15^N isotope-enriched membrane fragments have been obtained at 9.4 T (Figure 5(a,a′)) and 4.7 T (Figure 5(b,b′)). The spectra a and b were collected under the continuous illumination, while spectra a′ and b′ were obtained in the dark. In both magnetic fields, all signals are emissive. At 9.4 T (a), six signals at 259.6, 253.2, 249.3, 216.1, 193.2, and 189.0 ppm, respectively, are observed, whereas at 4.7 T (b), the signals at 249.3 and 216.1 ppm have been significantly decreased. This observation reflects the reduced influence of the TSM mechanism at 4.7 T, which relates to the electron acceptor. On the other hand, the DR mechanism becomes more active at lower fields [69]. Interestingly, due to the negative gyromagnetic ratio of ^15^N nuclei, the DR produces emissive signals. A comparable observation of emissive light-induced ^15^N signals has been reported from the carotenoid-less mutant *R. sphaeroides* R26, which is known to show a strong DR effect [52]. Such 1D spectra allow the assignment of the ^15^N chemical shift and are based on our preliminary studies on Chl *a* in PSI [53,54] and PSII [73], as well as BChl *a* in *R. sphaeroides* WT [74] and R26 [52]. The ^15^N chemical shift assignments are shown in Table 1. Our experimental data and magnetic shielding constants from DFT calculation show a good correlation with *R*^2^ = 0.9983 for the electron donor, and R^2^ = 0.9973 for the electron acceptor (see Appendix A), supporting the experimental assignments. 

From the chemical shift assignment, there is no hint for signal doubling, implying that both branches of HbRC are perfectly symmetrical and therefore equally active in light-induced ET. It appears that the functional symmetry break occurred later in the evolution of photosynthesis. 

### 2.4. The Effect of ^13^C Isotope Enrichment 

For obtaining connectivities in 2D ^13^C-^13^C NMR experiments, ^13^C enrichment is required. To this end, we adopted a strategy involving selective ^13^C isotope labelling [57,75] and isolation of membrane fragments [56]. While [4-^13^C] and [5-^13^C]-ALA-labelling patterns (Figure 3 in Section 1, shown in yellow and green, respectively) allow us to study the aromatic carbons, [3-^13^C]-ALA-labelling leads to isotope enrichment of the more peripheral carbon positions of the BChl *g*′ and 8^1^-OH-Chl *a_F_* macrocycles (Figure 3 in Section 1, shown in red). Since in HbRCs the cofactors are symmetrically arranged and the ET transfer occurs in parallel on two branches, the chemical shift assignment is relatively straightforward compared to the bacterial RC of *R. sphaeroides* WT having an asymmetric donor dimer leading to asymmetric electron transport [49,76]. Hence, backed by the ^15^N experiments, we expect to observe only a single set of donor signals as well as single set of acceptor signals. On the other hand, we will also look for whether signals are split, which would be a hint for small local symmetry breaks or disorder. 

Since the spectra will contain enhanced absorptive and emissive signals, we explore first the field dependence of the differently labelled samples with one-dimensional experiments (Figure 6A–C). Apparently, all three isotope labelling patterns ([4-^13^C]-ALA, [5-^13^C]-ALA, and [3,4,5-^13^C]-ALA) show the same trend: At 9.4 T, the emissive signals occur clearly, while the enhanced absorptive signals are weak. At 4.7 T the emissive signals remain strong, and the enhanced absorptive signals increase. According to the discussion above, the field dependence of the enhanced absorptive signals might be interpreted in terms of the DR mechanism which occurs selectively on donor nuclei. The overall pattern of field dependence observed in the three labelled samples is in line with that observed on the unlabelled sample (Figure 4). Hence, there is no evidence for an active role of the magnetic nuclei on the spin-chemical machinery. Compared to the signals observed from the unlabelled sample (Figure 4), the signals in Figure 6 appear to be broadened. The increase in linewidth can be interpreted as an effect of J-couplings among the labelled carbons. The signal at 112.6 ppm (*E*. 9.4 T) has a linewidth of 94 Hz in the [5-^13^C]-ALA-labelled sample and broadens to 144 Hz in the [3,4,5-^13^C]-ALA-labelled preparation. The signal at 190.8 ppm (*E*, 9.4 T) has a width of 57 Hz in the [4-^13^C]-ALA-labelled sample and of 116 Hz in the [3,4,5-^13^C]-ALA-labelled membrane. Moreover, in [3,4,5-^13^C]-ALA-labelled samples, spin diffusion between the carbons is very efficient, allowing for equilibrating signal intensities.

### 2.5. ^13^C Chemical Shift Assignment Based on One-Dimensional MAS NMR Spectra

The chemical shifts of the ^13^C atoms in BChl *g* and 8^1^-OH-Chl *a_F_* are still not reported in the literature; however, the chemical composition of the tetrapyrrole macrocycle of BChl *g* remains similar to that of BChl *a* [77] aside from the modifications at rings A and B. Furthermore, for 8^1^-OH-Chl *a_F_*, structural differences to Chl *a* are predominantly found at ring B [78] (see Appendix A). Therefore, the starting point of our assignment from one dimensional spectra is based on the chemical shifts of BChl *a* for BChl *g* and plant Chl *a* for Chl *a_F_* obtained from liquid [56,79] and solid-state NMR [56,80,81] in the literature. In addition, the magnetic shielding constants from DFT calculations provide support for the signal assignment.

Following the assumption that donor signals appear positive (see above), we make the working hypothesis that the positive (enhanced absorptive, *A*) signals are caused by the DR mechanism dominating over the TSM and originate from the BChl *g*’ donor, while the negative (emissive, *E*) peaks arise from the 8^1^-OH-Chl *a_F_* acceptor and are due to the TSM mechanism. Therefore, the positive signals will be studied at 4.7 T and negative signals at 9.4 T. The working hypothesis will now be evaluated by comparing experimental-and estimated chemical shifts based on DFT calculations (Table 1). 

The selectively [4-^13^C]-ALA-labelled cofactors (Figure 3 in Section 1, in yellow) carry the labelled carbons at the positions C1, C3, C6, C8, C11, C13, C17, and C19. The aliphatic signal of C17 appears at 52.8 ppm (*A*, at 4.7 T) and 53.9 ppm (*E*, 9.4 T) (Figure 6A, spectra a and b). Additionally, five more positive signals are shown at 169.6, 166.1, 155.2, 146.2, and 127.9 ppm and are assigned as C19, C6, C1, C11, and C13 of the electron donor, respectively, although the order of C6 and C19 is not definitively clear. There is also a weak positive signal at 133.6 ppm, which might originate from naturally abundant carbon and be attributed to C2 from the electron donor. Similarly, the small negative signal as the most deshielded carbon at 190.8 ppm is expected to be C13^1^ from the acceptor, which is also naturally abundant. The seven negative signals at 172.0, 157.3, 153.8, 147.0, 144.9, 140.3, and 127.3 ppm might originate from C19, C1, C6, C11, C8, C3, and C13, respectively, of the electron acceptor. Here, DFT calculations provide a good hint that the ^13^C magnetic shielding constant of C19 is more deshielded than that of C6 for both the electron donor and acceptor, leading to an enhanced linear coefficient (Table 1 and Appendix A).

The light-induced signals from [5-^13^C]-ALA membrane fragments are also obtained at 9.4 T (Spectrum a) and 4.7 T (Spectrum b) in Figure 6B. ^13^C isotope-enriched carbon positions are C4, C5, C9, C10, C14, C15, C16, and C20 (see Figure 3 in Section 1, in green). The positive signals at 162.3, 151.0, 148.8, and 147.6 can be assigned to C14, C16, C4 (or C9), and C9 (or C4), respectively, of the donor. Based on previous chemical shift assignments of BChl *a* [56], C9 is more deshielded than C4; however, the ^13^C magnetic shielding constants from DFT calculations show no obvious difference between C4 and C9 for the donor. The negative signals at 163.0, 157.8, 149.9, and 145.5 ppm can be assigned to C14, C1, C4, and C9, respectively, of the acceptor. For the acceptor, magnetic shielding constants from DFT calculations provide good agreement for all carbon positions. 

The assignment of the signals of methine carbons (around 100 ppm) is difficult since donor signals in this region also remain negative, as is known from RCs of *R. sphaeroides* R26 [82], probably due to a weak DR contribution at these positions. In total, eight signals are observed from methine carbons. The four negative signals at 112.6, 101.3, 108.9, and 102.6 ppm can be assigned to C10 and C15 of both the donor and the acceptor. The four signals at 97.2, 96.2, 93.2, and 92.0 ppm might arise from C5 and C20 of the donor and acceptor. From the study of the one-dimensional spectra, no hint for signal doubling is found. This is in line with the X-ray structure, which also implies that both halves of the HbRC are structurally identical. 

The photo-CIDNP MAS NMR spectra of the [3,4,5-^13^C]-ALA-labelled membrane fragments are shown in (Figure 6C). While in spectrum b (4.7 T), positive and negative signals heavily overlap, preventing interpretation, spectrum a (9.4 T) shows mainly negative signals arising from the acceptor. Since [3,4,5-^13^C]-ALA-labelling contains eight [3-^13^C]-ALA-labelled carbon positions including C2, C3^1^, C7, C8^1^, C12, C13^1^, C17^1^, and C18 (see Figure 3 in Section 1, in red), based on the previous study on [3-^13^C]-ALA-labelled cells of *Hb. mobilis* obtained at 4.7 T [57], we were able to differentiate the signals originating from the eight [3-^13^C]-ALA-labelled carbon positions. The positive signals from aliphatic carbons region, 29.9, 44.5, and 48.0 ppm, can be assigned to C17^1^, C7, and C18, respectively, and the signal at 114.9 ppm might originate from either C3^1^ or C8^1^ of the electron donor. Because the structural variations between BChl *g* and BChl *a* are particularly found at rings A and B, e.g., at the C3^1^ and C8^1^ positions, the chemical shifts of BChl *a* cannot be taken as a guideline for the assignment of these carbons. Thus, we rely on magnetic shieldings from the DFT calculation suggesting that the signal at 114.9 ppm originates from C8^1^ (see Table 1 and Appendix A).

The unexpectedly large positive intensities observed in the aliphatic carbons are noteworthy. Specifically, at 9.4 T, the weak positive signals from aromatic carbons, which makes intensity pumping hardly possible, might be due to the spectral overlap between positive and negative signals. Consequently, we hypothesize that the strong positive signals, contributed by the DR mechanism, such as C4, C6, C16, and C19 of aromatic carbon, induced an intensity enhancement in neighboring aliphatic carbons, notably C7, C17, C17^1^, and C18, resulting in significantly higher intensity. 

The most deshielded negative signals at 190.8 ppm arise from C13^1^ the acceptor undoubtedly. The other negative signals arising from the electron acceptor of aromatic carbons, 136.8 and 134.6 might arise from C2 and C12. Magnetic shieldings from DFT calculation support C2 being more deshielded than C12 (see Table 1 and Appendix A). In the aliphatic region, the negative signal at 64.9 ppm, 49.8 ppm, and 31.7 ppm can straightforwardly be assigned to C8^1^, C18, and C17^1^, respectively, of the acceptor. 

From the photo-CIDNP 1D spectra of [4-^13^C]-, [5-^13^C]-, and [3,4,5-^13^C]-ALA-labelled cofactors measured at different magnetic field, 4.7 T and 9.4 T, we could assign 14 aromatic carbons and 6 aliphatic carbon positions for both the electron donor and acceptor, respectively. Our hypothesis, which asserts that positive signals result from the electron donor, while negative signals originate from the electron acceptor, has been confirmed through a strong consistency between the calculated magnetic shielding constants and the experimental chemical shifts. Interestingly, most of the carbon signals arising from the electron donor are more shielded than those from the electron acceptor except for methine carbons. 

The chemical shift assignment based on 1D spectra, however, has its limits: the signal doubling of methine carbons still presents ambiguity since the assignment to either the donor or acceptor remains unclear. This ambiguity arises from the fact that all the signals between 100 and 90 ppm are negative in [5-^13^C]-ALA at both 4.7 T and 9.4 T, which can be assigned to C5 and C20 of the donor. These signals turn positive upon [3,4,5-^13^C]-ALA- labelling. One might speculate that isotope labelling strengthens the DR mechanism; however, spin diffusion from nearby nuclei having strong positive intensity might provide a more convincing explanation. Furthermore, we have not identified any noticeable positive signals emerging from carbon C3^1^ of the donor, implying low electron spin density in the donor triplet state. 

### 2.6. ^13^C Chemical Shift Assignment Based on Two-Dimensional MAS NMR Spectra

The 2D ^13^C-^13^C photo-CIDNP DARR MAS NMR spectra of [3,4,5-^13^C]-ALA-labelled membrane fragments allow for conclusive ^13^C assignments of both the electron donor and the acceptor cofactors, which remained ambiguous from the analysis of the 1D light-induced MAS NMR spectra of selectively labelled samples (see above). The proton-driven spin diffusion efficiency depends on spectral overlap [83,84] as well as local rigidity [85,86]. To optimize the occurrence of connectivities, the spin diffusion mixing time was carefully optimized to 50 ms. The 2D ^13^C-^13^C photo-CIDNP DARR MAS NMR spectra (Figure 7B,C) allow connectivities to be identified. In the spectrum in Figure 7C, measured at 4.7 T, we observe positive and negative diagonal and cross-correlation peaks from both the electron donor and acceptor. We were able to identify 16 positive cross peaks and 9 negative cross peaks. The 2D spectrum in Figure 7B, obtained at 9.4 T, allows for identification of 23 negative and 6 positive cross peaks. Following the discussion of the 1D data (see above, see Figure 7A,D), all positive signals originate from the electron donor and are caused by the DR mechanism dominating over the TSM. The negative signals caused by the TSM originate from the acceptor. Solely in the region of the methine carbons (120 to 80 ppm), donor signals might appear negative, too, although in [3,4,5-^13^C]-ALA-labelled samples, spin diffusion might lead to an inversion of the sign. 

For the sake of clarity, in the following discussion of assignments, carbons associated with the electron donor are designated as “P”, while carbons belonging to the electron acceptor are denoted as “A_0_”. First, we discuss the positive signals in the spectrum obtained at 4.7 T (Figure 7C). The aliphatic carbons are taken as the starting points for the chemical shift assignment since the magnetization transfer is mediated by ^1^H nuclei in the ^13^C-^13^C DARR experiment, leading to strong signals for the aliphatic carbons [81,87]. 

Thus, starting from C17^1^P at 29.9 ppm, there are the two correlations to signals at 52.8 and 148.8 ppm. While the signal at 52.8 ppm can straightforwardly be assigned to C17P, the signal at 148.8 ppm belongs to the carbons detected upon [5-^13^C]-ALA-labelling and could not be decided between C4P and C9P on the basis of 1D spectra (see above). According to the 3D structure of the donor dimer (PDB entry 5v8k) [21], the intermolecular through-space distances from C17P and C17^1^P to C4P are 6.1 and 6.2 Å, while the distance to C9P would be 7.2 and 8.1 Å (see Appendix A). Therefore, an assignment of the signal at 148.8 ppm to C4P would be more plausible. Thus, the signal at 147.6 ppm possibly originates from C9P. We also assign the signal at 169.6 ppm to C19P since it is correlated with the aliphatic carbon C18P (48.0 ppm). Moreover, we assign the signal at 166.1 ppm to C6P since there is a strong correlation with C7P at 44.5 ppm. And the positive signal at 114.9 ppm is conclusively assigned to C8^1^P rather than C3^1^P, based on its correlation partners C4P and C8P.

In the same way, starting from negative aliphatic carbons signals as C17^1^A_0_, C18A_0_, and C17A_0_, the negative cross-correlation signals will be assigned (Figure 7B). We can conclude that the negative signal at 172.0 ppm originates from C19A_0_ based on the cross peak C18A_0_/C19A_0_. The negative signal at 157.3 ppm, also observed in the [4-^13^C]-ALA-labelled sample, is assigned to C1A_0_ due to the correlation with the nearby C17A_0_ and C17^1^A_0_. The negative signals at 134.6 and 136.8 ppm can be clearly distinguished: The signal at 134.6 ppm is assigned to C12A_0_ due to the correlation with the carbons C10A_0_, C11A_0_, and C13^1^A_0_. Hence, the signal at 136.8 ppm is assigned to C2A_0_, and there is a weak cross peak with C1A_0_.

The ^13^C-^13^C photo-CIDNP DARR MAS NMR spectra show the eight signals of the methine carbons in the region from 120 ppm to 80 ppm. One of these signals that is expected for C20P (at 92.0 ppm) has a positive cross peak with C7P as well as with C6P in the [3,4,5-^13^C]-ALA sample (Figure 7C). Concerning carbon C5P (97.2 ppm, *A*), it has a positive correlation with C4P at 148.8 ppm, representing it as an electron donor. In the ^13^C-^13^C DARR photo-CIDNP MAS NMR data acquired at 4.7 T with a short mixing time of 2.5 ms (see Appendix A), it became possible to differentiate between the C5 carbons of the donor and the acceptor. This distinction was achieved by identifying substantial correlations between C5P (at 97.2 ppm, *A*) and C1P (at 155.2 ppm, *A*), as well as between 5A_0_ (at 96.2 ppm, *A*) and 9A_0_ (at 145.5 ppm, *E*), even though these correlations exhibit a positive sign. Neither C10P (101.3 ppm, *E*) nor C15P (at 108.9 ppm, *E*) have correlation peaks at both 4.7 T and 9.4 T, although their negative diagonal peaks appear. It might be that the cross-signals with an electron donor exhibiting a positive phase could converge to zero. Therefore, it is probable that these signals originate from the electron donor.

The signal of C20A_0_ (at 93.2 ppm, *E*) shows a negative correlation with C1A_0_ at both 4.7 T and 9.4 T. Similarly, the signal at 112.6 ppm, here assigned to C10A_0_, has strong correlations with C12A_0_ (134.6 ppm) as well as with C9A_0_ (145.5 ppm) at 4.7 T. The observed spectral trend persists at a field strength of 9.4 T as well. Strong emissive correlations of C10A_0_ (at 112.6 ppm, *E*) with C12A_0_, and C11 A_0_ as well as C13^1^A_0_ are observed. Additionally, the signal for C15A_0_ (at 102.6 ppm, *E*) also shows a negative cross peak with C14A_0_ at 9.4 T; thus, it is assigned to the electron acceptor. The observation of C10A_0_ and C15A_0_ provides empirical support for our hypothesis, as discussed above, that C10P (101.3 ppm, *E*) and C15P (at 108.9 ppm, *E*) originate from the electron donor. 

Consequently, by taking into account the sign differences in cross peaks and the presence of positive or negative correlations, we confidently assigned carbons to either electron donors or acceptors. This clarification is particularly valuable for carbons that initially remained ambiguously characterized by the analysis of 1D spectra. The sign change in [3,4,5-^13^C]-ALA, notably in the case of methine carbons, is remarkable. Nevertheless, their assignment is determined based on their correlation partners in the 2D DARR photo-CIDNP MAS NMR spectra.

We also note that the 2D ^13^C-^13^C correlation spectra do not provide many cross peaks which arise from neighboring carbon pairs (see Figure 7B,C). The intensity of the diagonal peaks induced by the solid-state photo-CIDNP effect could indeed be the main factor influencing the strength of the cross peaks. This would explain why no cross peak is observed in subsequent correlations, such as C1P/C2P, as the diagonal signals from these nuclei have a much lower intensity in the 1D spectrum. However, signals of nuclei such as C4P and C17P show the highest intensity, especially at 4.7 T. Consequently, the cross peaks appear even at a mixing time of 2.5 ms (Appendix A), which facilitates polarization transfer over long distances. 

Additionally, in Spectrum C, obtained from 4.7 T, we observe two cases of signal splitting: at 187.8 and 188.0 ppm (59.4 Hz) for C13^1^P as well as at 189.9 and 191.1 ppm (63.9 Hz) for C13^1^A_0_. This phenomenon, which was also discussed earlier in the 1D spectrum of the [3,4,5-^13^C]-ALA-labelled sample, can be rationalized in terms of J-coupling interactions. The amount of splitting, induced by 13^1^P/13P and 13^1^P/14P at the donor as well as 13^1^A_0_/13A_0_ and 13^1^A_0_/14A_0_ for acceptor, is typical for aromatic carbon systems [88]. Furthermore, we observe a subtle signal doubling at 127.3 and 126.3 ppm (54 Hz) for C13A_0_, providing evidence for a neighboring J-coupling interaction with C13^1^A_0_. This phenomenon was not evidently observed at 9.4 T; however, a weak splitting (about 60 Hz) of C13^1^P is consistent with the observation at 4.7 T. Hence, signal doubling in [3,4,5-^13^C]-ALA-labelled samples at low field is due to J-coupling and does not necessarily imply conformational heterogeneity.

The conclusive assignments of ^13^C based on our 1D photo-CIDNP and 2D DARR photo-CINDP MAS NMR experiments are shown in Table 1. The correlation between the experimentally obtained ^13^C chemical shifts and the calculated ^13^C magnetic shielding constants shows good agreement with R^2^ = 0.9948 for the electron donor, BChl *g*′, and with R^2^ = 0.9923 for the electron acceptor, 8^1^-OH-Chl *a_F_* (Appendix A), further substantiating these assignments. 

### 2.7. Long-Range Transfer of Nuclear Hyperpolarization 

The DARR spectra with a mixing time of 50 ms (Figure 7) of [3,4,5-^13^C]-ALA-labelled cofactors also reveal correlation peaks caused by long-range transfer of hyperpolarization up to ca. 6 Å (the correlation pairs from electron donor and acceptor, with distance between atoms based on the X-ray structure, are listed in Appendix A). That is in line with previous reports on [3-^13^C]-ALA-labelled RC of *R. sphaeroides* WT with a mixing time of 2 s [49]. The transfer of hyperpolarization among ^13^C atoms is mediated by spin diffusion operating through space and allowing for intermolecular contacts. Especially the [3,4,5-^13^C]-ALA-labelling pattern allows for efficient spin diffusion along the aromatic systems. Consequently, long-distance transfer can also occur within a mixing time of 50 ms. 

Furthermore, there are multiple intermolecular cross peaks occurring within the donor special pair. The interplanar distance between both macrocycles is approximately 3.1 Å, and rings A and B are partially overlapping. Such intermolecular cross-correlations identified are C7P/C20P (distance = 6.5 Å), C4P/C17^1^P (6.2 Å), C4P/C17P (6.1 Å), and C8^1^P/C4P (5.5 Å) (see Appendix A). In particular, the transfer between C7P and C20P might involve a multi-step process since the correlation of C7P/C6P and C6P/C20P are observed: Initially, the polarization might be transferred from C7P to C6P, with a distance of 1.5 Å, occurring through intramolecular interactions. Subsequently, this polarization is relayed to C20P, which is positioned at a distance of 5.5 Å, through intermolecular interactions between the two donor cofactors comprising a special pair. 

When examining the correlations within the electron acceptor, we have also identified long-distance correlations, such as C15A_0_/C6A_0_ (6.3 Å) and C3A_0_/C13A_0_ (8.4 Å). However, in both cases, multi-step polarization might be considered due to the correlations with intermediate carbons (see Appendix A). The correlation between 15A_0_ and 6A_0_ may involve several steps due to the correlations of 15A_0_/14A_0_, 14A_0_/12A_0_, 12A_0_/10A_0_, and 10 A_0_/6A_0_. Likewise, the transfer over the distance of 8.4 Å between 3A_0_ and 13A_0_ might involve a sequence of steps, as suggested by the observed correlations: 3A_0_/2A_0_, 2A_0_/1A_0_, 1A_0_/17A_0_, 17A_0_/17^1^A_0_, 17^1^A_0_/13^1^A_0_, and 13^1^A_0_/13A_0_. However, in this case, the spectral overlap may also be the reason inducing such a long-distance correlation due to the close chemical shift (3A_0_: 140.3 ppm, 13 A_0_: 127.3 ppm). This observation is akin to findings from previous research on the *R. sphaeroides* WT [49], where long-range connectivities were explained by multi-step transfer processes, involving a maximum of six steps leading to correlations with a distance of 13.1 Å between the intermolecular donor special pair, P_L_, and P_M_.

At the acceptor site, carbon C13^1^A_0_ shows five correlations with C10A_0_, C11A_0_, C12A_0_, C13A_0_, C14A_0_, and C17^1^A_0_ at 9.4 T. Similarly, carbon C17^1^A_0_ shows correlations with C1A_0_, C13^1^A_0_, C17A_0_, and C18A_0_. Efficient spin diffusion might be related to the proton-rich environment in the close proximity (e.g., surrounding amino acids such as Arg-554, Ser-553, hydrogen bonding between C=O of C13^1^A_0_ and Arg-406 as well as Ser-553 [21]).

### 2.8. The Relation of Chemical Shifts and Redox Potential 

Chemical shifts, reflecting the electronic shielding of nuclei, report on the electron density and are, therefore, related to electronic and electrochemical properties. Based on our experimental chemical shift assignment, we calculated the sum of the aromatic chemical shifts (SACS) of the primary electron donor and acceptor of HbRC and compared these values with other photosynthetic RCs from purple bacteria, and PSII. The experimental chemical shift values for RC from heliobacteria *Hb. mobilis* were obtained in the present study, while previous photo-CIDNP MAS studies provided data specifically for RCs of purple bacteria *R. sphaeroides* WT [89,90,91] and of PSII from spinach [92,93] (chemical shifts are listed in Appendix A). In case of the special pair of *R. sphaeroides* WT, the averaged value of P_L_ and P_M_ has been used for the calculation since they act as a supermolecule as a dimer of the primary donor.

The sum of the aromatic chemical shifts (SACS) in the electron donor follows a trend among the RCs: heliobacteria < purple bacteria < PSII, whereas the electron acceptor exhibits a different order: PSII < purple bacteria < heliobacteria (see the Table 2). Interestingly, the SACS values of the donor and acceptor show a good linear correlation (R^2^ value of 0.9530, see Figure 8A) among three photosynthetic RCs, suggesting a potential spectroscopic method for estimating relative redox potentials in ET systems. 

We also find that the sum of all chemical shifts of the aromatic carbons of the electron donor of the HbRC (for details, see Appendix A) is around 25 ppm lower (i.e., more shielded) than that of the acceptor, implying a significant gradient of electron density in the electronic ground state. This is corroborated by the sum of absolute shielding constants from our DFT calculations, as the donor with 647.4 ppm as the sum of shielding constants exhibits a larger value than the acceptor with 622.5 ppm. In the following, we refer to this difference as the “overall chemical shift difference” (OCSD), reflecting how these cofactors experience either shielding or deshieding within their protein environment. Notably, the OCSD of PSII is −51.5 ppm, implying that the most deshielded electron donor is located within this plant RC, while the RCs of purple bacteria *R. sphaeroides* WT falls between Heliobacteria and PSII (see Table 2). 

While electrostatic interactions and molecular properties have been explored for charge separation and ET among different RCs in photosynthetic systems [96,100,101,102,103], the relationship between chemical shifts and redox properties in the ground state of active cofactors has not yet been discussed. Here, our investigation suggests such a connection on the basis of these three types of photosynthetic RCs of heliobacteria, purple bacteria *R. sphaeroides* WT, and PSII. We find a strong linear correlation between the SACS values of donors and their redox potentials (R^2^ value of 0.9831, see Figure 8B). Hence, the electron donor of PSII, P680, allowing for water oxidation in oxygenic photosynthesis, exhibits the most deshielded properties. Conversely, the HbRC, having the lowest redox potential, has the most shielded donor. Additionally, our data revealed a linear relationship between the SACS of the electron acceptors and their redox potentials (correlation coefficient of R^2^ = 0.9612, see Figure 8C), representing the most deshielded acceptor cofactor having lowest redox potentials. This further strengthens our understanding of how redox potentials relate to the chemical behavior of the acceptor molecule. 

Our findings demonstrate that the electric potential of an ET system might be linked to the SACSs value which would, vice versa, imply a straightforward method to obtain information of the redox properties of the active cofactors in their electronic ground state. It can also shed light on how nature utilizes specific cofactors, adjusting them within the protein environment to create various properties crucial for their function. 

### 2.9. Electron Spin Density Distribution 

Figure 9 shows the electron spin density maps of donor and acceptor cofactors reconstructed based on ^13^C and ^15^N NMR intensities enhanced by the solid-state photo-CIDNP effect. Since spin diffusion in selectively ^13^C labelled samples would modify the original intensity, intensities obtained from samples at natural abundance are used for analysis. For the strength of the ^15^N solid-state photo-CIDNP effect, samples can be isotope-enriched since spin diffusion between nitrogen nuclei is negligible. 

In Figure 9, the size of the spheres has been normalized to the most intense signal arising from C9 (at 147.6 ppm, *A*) for ^13^C and N22 (at 259.6 ppm, *E*) for the ^15^N nuclei. The ^13^C intensities of the donor (Figure 9A) are positive and caused by dominance of the DR over the TSM. Solely the methine carbon position appears negative, implying a weak DR and lower electron spin density in the donor triplet state. The ^15^N intensities of the four pyrrole nitrogens caused by the DR appear negative (Figure 9B). Spin populations from Mulliken population analysis provide a reasonable estimation of the spin density distribution in the neutral donor triplet, which generally agrees with the experimental data (correlation coefficient of 0.6983 for ^13^C; see Appendix A). The calculations also confirm that the electron spin density on the methine carbons is low. The maximum electron spin density distribution in the triplet state is observed in the slightly overlapping region of the dimer (i.e., ring B). Aromatic amino acid residues in close proximity to ring B, such as Phe-511, might further stabilize electron spin density (see Appendix A). Surprisingly, in the central overlapping region (i.e., ring A), the spin density appears to be relatively small.

The trend is different from PSII observed in spinach [73], showing the maximum of local spin density on the carbons of pyrrole ring C including methine bridge C15. That shift of spin density has been explained by the hinge model with a tilted axial histidine residue leading to an electronic interaction between the donor Chl and the axial histidine, contributing to an increase in the redox potential [73,104,105]. Thus, the conformation of the cofactors and the effect of the protein matrix appear to be conserved in the photosynthetic RCs of the various organisms, which might allow its function to be optimized under specific conditions. 

Figure 9C,D represent the local electron spin density distribution observed on ^13^C and ^15^N nuclei, respectively, on the electron acceptor in its radical-pair anion state. The TSM mechanism requires hyperfine anisotropy Δ*A* and has been demonstrated to describe the results of steady-state photo-CIDNP MAS experiments [50]. The polarization observed on the cofactors in RCs of *R. sphaeroides* WT is roughly correlated to Δ*A*^2^ [82]. Since these intensities are due to the TSM, we compare the experimental intensities with the calculated square of anisotropic hyperfine parameter, Δ*A*^2^ (see Appendix A). The correlation coefficient, R^2^ = 0.7745 for ^13^C, supports the interpretation that these signals originate from the TSM. No meaningful correlation for ^15^N, however, was found. 

The electron acceptor cofactor has not yet been studied in detail in PSI and PSII by photo-CIDNP MAS NMR. In HbRCs, however, the ^13^C photo-CIDNP intensity pattern reveals pronounced electron spin delocalization on pyrrole rings C and E as well as on the methine bridge C10. The asymmetry of the electronic structure may be influenced by the protein matrix. Amino acid residues in close proximity are Arg-406, Arg-554, and Ser-553. In particular, Arg-406 and Ser-553 potentially form a hydrogen bond with the C13^1^ keto-oxygen of the acceptor [21] (see Appendix A). Interestingly, the matrix tuning the acceptor cofactor also affects its redox potential: as mutagenesis studies demonstrate, the redox potential increases by surrounding two arginine residues, Arg-406 and Arg-554, due to the hydrogen bonding and hydrophobicity [39]. Additionally, the localization of the electron spin density on pyrrole ring C might facilitate ET towards the nearby terminal acceptor Fx (see Appendix A). Thus, our observations of the localization of the electron spin might provide a key for the mechanistic understanding of the high efficiency of ET in natural photosynthetic RCs.

## 3. Materials and Methods

### 3.1. Sample Preparation

Cells of *Hb. mobilis* strain DSMZ 6151 were used in this study [56,57,58]. The cells were cultured in medium no. 1552 [106] anaerobically at 37 °C under continuous light with 2000 lux for 7–9 days. For the selective ^13^C isotope labelling, cells were cultured in the medium containing 1.5 mM [4-^13^C]-*δ*-aminolevulinic acid (ALA), [5-^13^C]-ALA, or [3,4,5-^13^C]-ALA and harvested by centrifugation (5000 rpm). ^13^C-ALA-labelling patterns are described in Figure 3 in Section 1. 

For the ^15^N isotope labelling, cells were cultured in the medium containing 62 mM ^15^NH_4_Cl [58]. The grown cells were harvested and uniformly suspended in deoxygenated 20 mM Tris**･**HCl buffer (pH 8.0) containing 10 mM of sodium ascorbate. A small amount of DNase was added, and the cell was lysed by sonication for 30 min. Afterwards, the lysate was centrifuged at 20,000× *g* for 15 min at 4 °C to remove broken cells and debris. The resulting supernatant was ultracentrifuged at 200,000× *g* at 4 °C for 4 h. The obtained pellet containing the membrane fragments was resuspended in 20 mM of tris buffer containing 10 mM of sodium ascorbate with 0.02% sulfobetaine-12 (SB-12) detergent (pH 8.0). The sample was reduced with 0.1 M of sodium dithionite under nitrogen gas flow and packed in 3.2 mm and 4.0 mm transparent sapphire rotors for photo-CIDNP MAS NMR experiments, respectively.

### 3.2. NMR Measurement

#### 3.2.1. NMR Set Up

NMR experiments have been performed with a 3.2 mm and 4 mm double-resonance MAS probe using a Bruker ACANCE NEO and AVANCE III spectrometer (Bruker Biospin GmbH, Karlsruhe, Germany), respectively. These spectrometers operated at a proton Larmor frequency of 400.15 MHz (9.4 T) and 200.13 MHz (4.7 T). For illumination, a 488 nm 1 W continuous-wave laser (Genesis MX488-1000 STM OPS Laser-Diode System, Coherent Europe B.V., Zeist, The Netherlands) was used. 

The probes were equipped with a light fiber which is connected to a MAS stator for illumination of the sample during the measurement [107]. A transparent sapphire rotor packed with membrane fragments was frozen in the dark at a slow spinning frequency of ca. 400 Hz to ensure a homogenous sample distribution on the wall [108]. The experiment was conducted at a temperature of 235 K and a spinning frequency of 8 kHz. All ^13^C NMR spectra were referenced to the ^13^COOH response of solid *L* tyrosine**･**HCl at 172.1 ppm, while ^15^N NMR spectra were calibrated using amino NH_2_ signal of histidine**･**HCl at 49.09 ppm. 

#### 3.2.2. One-Dimensional ^13^C and ^15^N Photo-CIDNP MAS NMR 

Both dark and photo-CIDNP spectra were obtained with a Hahn-echo pulse sequence [107]. For the recycle delay for ^13^C, a period of 4 s, and for ^15^N, 6 s was used. The π/2 ^13^C pulse was applied at a radio-frequency (rf) field strength of 67 kHz at 4.7 T and of 80 kHz at 9.4 T, respectively. The π/2 ^15^N pulse was applied at an rf field strength of 32 kHz at 4.7 T and 50 kHz at 9.4 T. 

Swept-frequency two-pulse phase-modulation (SW_f_-TPPM) heteronuclear decoupling [109] with 61 kHz at 4.7 T and 81 kHz at 9.4 T, respectively, was used during the acquisition. Artificial line broadening of 30 Hz was applied prior to Fourier transformation for all NMR spectra.

#### 3.2.3. Two-Dimensional ^13^C-^13^C Photo-CIDNP DARR MAS NMR under Continuous Illumination

Two-dimensional ^13^C-^13^C photo-CIDNP DARR spectra were acquired with an optimized mixing time of 50 ms. Throughout the mixing period, continuous-wave decoupling at ^1^H was used to fulfill the rotary resonance conditions ν_1_ = *n*ν_R_ (where *n* = 1 or 2) [83,84] with *n* = +2 at an rf field strength of 16 kHz. The total number of recorded scans was 512 with 120 increments in the indirect dimensions, and a relaxation delay time of 4 s was used for both magnetic fields. The light-induced 2D ^13^C DARR spectrum was acquired over a period of approximately 3 days (58 h). Zero filling to 4 K and an exponential apodization of 50 Hz was applied prior to Fourier transformation. Frequency discrimination was achieved using the States-TPPI method [110]. 

### 3.3. DFT Calculation 

Structures of the cofactors BChl *g*’ and 8^1^-OH Chl *a_F_* were taken from the crystal structure of the homodimeric RC of *H. modesticaldum* (pdb code: 5V8K [21]). Using Avogadro [111], missing hydrogen atoms were added, and the longer sidechain of each cofactor was truncated to a methyl group after the ester bond. Before the actual property calculations, optimizations were performed with the PBEh-3c method [112], and subsequent frequency calculations with the same level of theory confirmed that the found stationary points are minima on the potential energy surface. 

As the experimentally observed chemical shifts originate from the neutral singlet ground state of the cofactors, corresponding calculations were realized for both the donor and the acceptor. For the determination of the NMR properties, the PBE0 [113,114] functional was employed together with the pcSseg-2 basis set [115], which was optimized for the calculation of nuclear magnetic shielding constants. We used the RIJCOSX approximation together with default auxiliary basis sets and required very tight SCF convergence. The calculated magnetic shielding constants were plotted against the measured chemical shifts. Linear regression then yielded the parameters for estimating the chemical shifts based on magnetic shielding constants from DFT calculations [116,117]. These predicted values based on a linear fit are shown together with their experimental counterparts in Table 1 and Appendix A. 

As for the electron acceptor, it is expected that the relative photo-CIDNP intensity roughly correlates with the square of the anisotropy of the hyperfine coupling tensor Δ*A*^2^; we calculated hyperfine coupling constants for the radical anion state of 8^1^-OH Chl *a_F_*. After geometry optimization with PBEh-3c for this state, we employed the TPSSh [118,119] functional together with the pcH-2 basis set [120], which is optimized for the calculation of hyperfine coupling constants. As for the NMR calculations, very tight SCF convergence was required; however, a part of that program’s defaults was used. The principal components of the hyperfine coupling tensors were extracted from this calculation, and the anisotropy was determined as Δ*A*= *A_zz_* − (*A_xx_* + *A_yy_*)/2. These results can be found in Appendix A.

In the case of the electron donor, a correlation between relative photo-CIDNP intensity and spin density of the neutral triplet state of the donor is expected. Therefore, we performed calculations for BChl *g*’ in this state with the same levels of theory as used for the electron acceptor. The spin populations obtained from Mulliken population analysis were then extracted from the TPSSh/pcH-2 calculation and are reported in Appendix A.

The calculations for the neutral singlet ground states of both cofactors were conducted with Orca version 4.2.1 [121], whereas version 5.0.3 [122] was used for the calculations of the acceptor in its radical anion state and of the donor in the neutral triplet state.

## 4. Conclusions

HbRCs are ancient photosynthetic RCs showing the solid-state photo-CIDNP effect. The analysis of the effect is, compared to other RCs, unequivocal since:These RCs are homodimeric, i.e., only two cofactors appear in the NMR spectra.The lifetime of the SCRP is similar in both spin-states, singlet, and triplet. That implies that the DD mechanism is negligible, and the number of spin-dynamical mechanisms is reduced to two: the DR and TSM.All positive ^13^C signals originate from the donor caused by the dominance of the DR over the TSM (except for methine carbons). The TSM mechanism causes negative signals originating from the acceptor cofactor.In the ^15^N photo-CIDNP MAS NMR spectra, all signals are negative since both DR and TSM produce negative intensities, implying that, due to the negative gyromagnetic ratio of ^15^N spin sorting during the coherent singlet–triplet interconversion, this changes the sign.The field dependence of DR and TSM provides a simple tool to optimize the observation of either the donor or acceptor cofactor.

Based on this, complete sets of chemical shifts from both donor and acceptor cofactors have been disentangled. The data imply that the aromatic system of the donor macrocycle is in sum 25 ppm more shielded than that of the acceptor. This difference of the sum of chemical shifts between donor and acceptor might provide a simple measure of the redox properties of ET systems. 

The electron spin density map of the donor triplet state, reported by DR intensities, exhibits a maximum at pyrrole ring B, possibly influenced or preserved by amino acids in close proximity. Conversely, the electron spin density map of the radical-pair anion state of the electron acceptor, encoded by TSM intensities, demonstrates notable delocalization on ring C, pointing towards the terminal acceptor Fx. 

Structural and electronic information reported here might provide guidelines for the synthesis of artificial photosynthetic systems. 

## Figures and Tables

**Figure 1 molecules-29-01021-f001:**
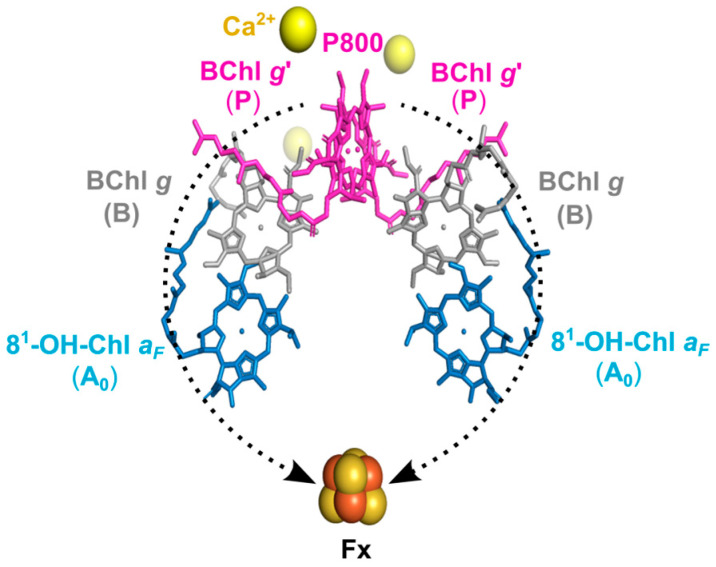
Arrangement of the cofactors in the HbRC of *H. modesticaldum* as shown by X-ray crystal structure of an RC [PDB entry 5V8K] [21]. The ET chain in HbRC consists of the special pair, P800, formed by two BChl *g*′ cofactors serving as the primary electron donor; the two accessory cofactors, B and BChl *g*; as well as two electron acceptor molecules, A_0_ and 8^1^-OH-Chl *a_F_*. Furthermore, the terminal electron acceptor is the 4Fe-4S cluster Fx. ET takes place on both branches (the electron pathways are shown with dashed black arrow). There is no carotenoid near the special-pair donor which would allow for fast relaxation of molecular triplet states.

**Figure 2 molecules-29-01021-f002:**
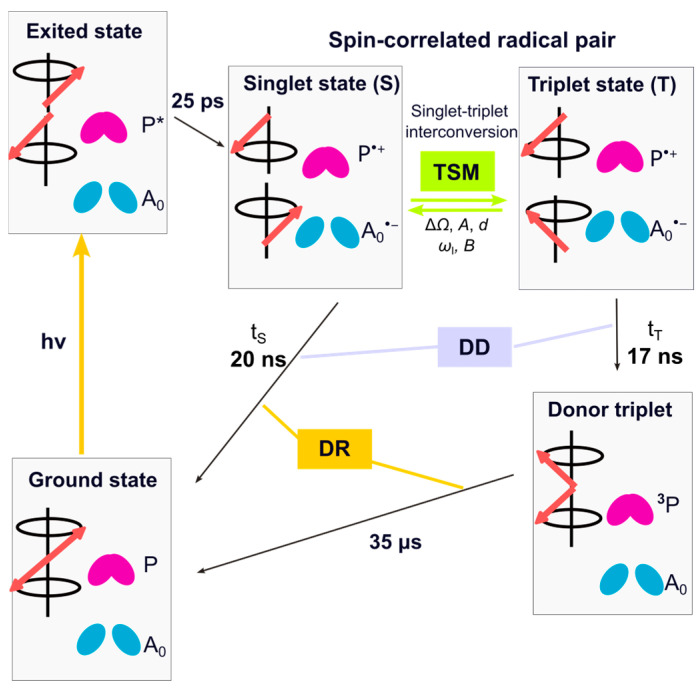
Kinetics and spin dynamics of cyclic ET in pre-reduced HbRCs. The red arrows represent the vector of electron spin states. Upon light absorption, the primary electron donor undergoes electronic excitation to form the excited state P*. Within 25 ps, ET from P* to the primary electron acceptor A_0_ occurs, resulting in the formation of a radical pair in a pure singlet state. Due to hyperfine interactions with the nuclei and the difference in *g*-value, Δ*g*, the radical pair evolves between the singlet and triplet state. During the spin evolution, nuclear hyperpolarization is generated via three-spin mixing (TSM). While the radical pair is spin-allowed to recombine during its singlet state, for the triplet state, a direct recombination is spin-forbidden, and, therefore, a donor triplet (^3^P) is formed. Additionally, due to the different kinetics of two decay channels, the contribution of the differential decay (DD) mechanism to nuclear spin hyperpolarization occurs. In HbRCs, the difference in lifetime of these two branches (t_S_ = 20 ns vs. t_T_ = 17 ns) [40,41] is negligible, allowing the DD contribution in the present system to be disregarded. In HbRCs, the donor triplet lifetime (^3^P) is ~35 µs [14,23,41,42,43], causing the occurrence of the differential relaxation (DR) mechanism.

**Figure 3 molecules-29-01021-f003:**
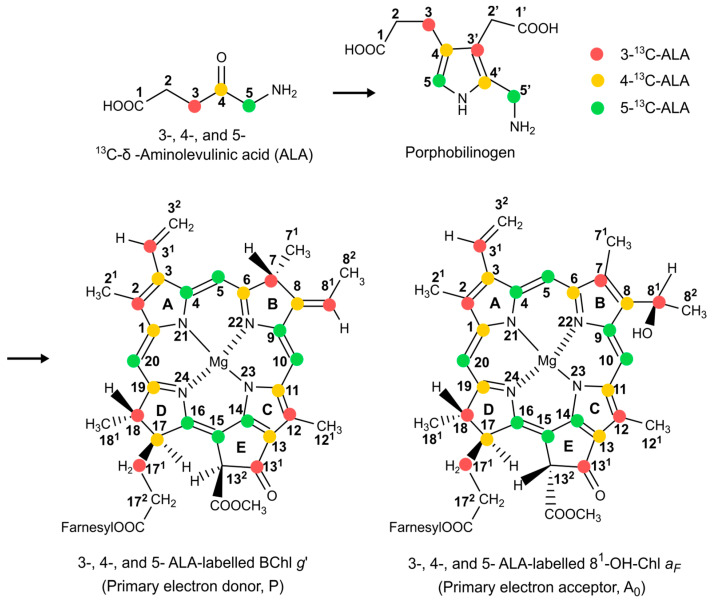
The biosynthetic pathway for the formation of selective ^13^C labelled BChl *g*’ and 8^1^-OH-Chl *a_F_* by feeding 3-, 4-, and 5-ALA in red, yellow, and green, respectively. Numbering is according to IUPAC nomenclature.

**Figure 4 molecules-29-01021-f004:**
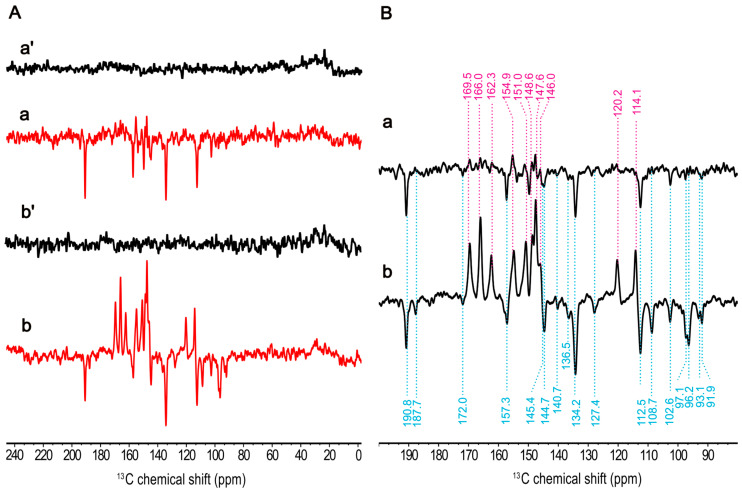
^13^C photo-CIDNP MAS NMR spectra from natural abundant membrane fragments from *Hb. mobilis* obtained at 9.4 T in the dark (**a′**) and under the illumination with 488 nm (**a**) as well as at 4.7 T in dark (**b′**) and under the illumination (**b**) in (**A**) for ca. 10 h. A cycle delay of 4 s at 9.4 T and of 2 s at 4.7 T was used. A MAS frequency of 8 kHz was employed for the experiments conducted at 235 K. Expanded light-induced spectra acquired at 9.4 T (**a**) and 4.7 T (**b**) in (**B**).

**Figure 5 molecules-29-01021-f005:**
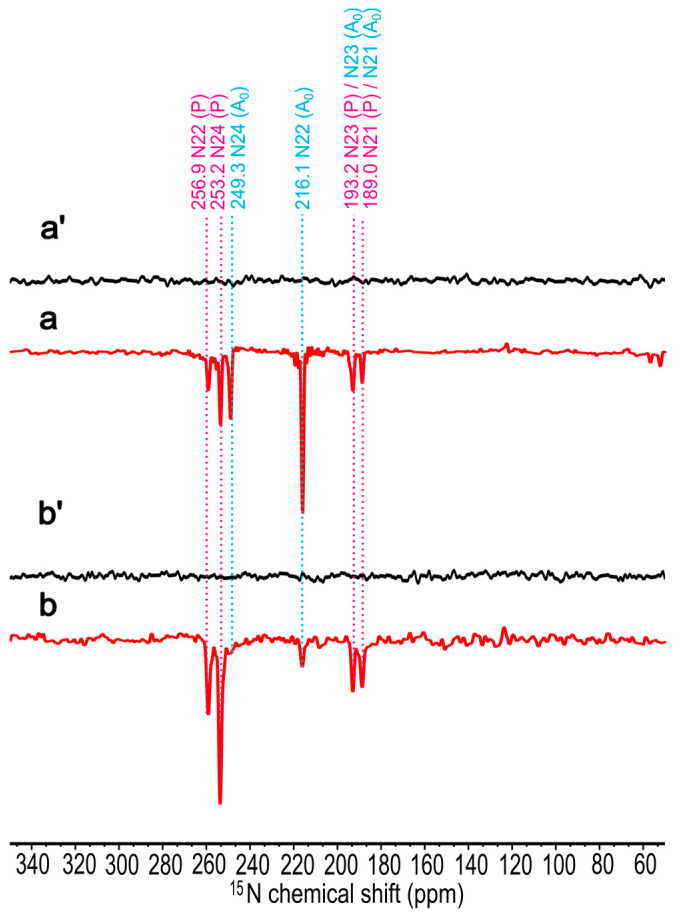
The ^15^N photo-CIDNP MAS NMR spectra of uniformly ^15^N isotope-enriched membrane fragments of *Hb. mobilis* obtained at 9.4 T in the dark (**a′**) and under the continuous illumination (**a**) as well as at 4.7 T in the dark (**b′**) and under the continuous illumination (**b**). For illumination, laser excitation at 488 nm was used. All experiments were recorded for 10 h at a MAS frequency of 8 kHz, a cycle delay of 6 s, and a temperature of 235 K.

**Figure 6 molecules-29-01021-f006:**
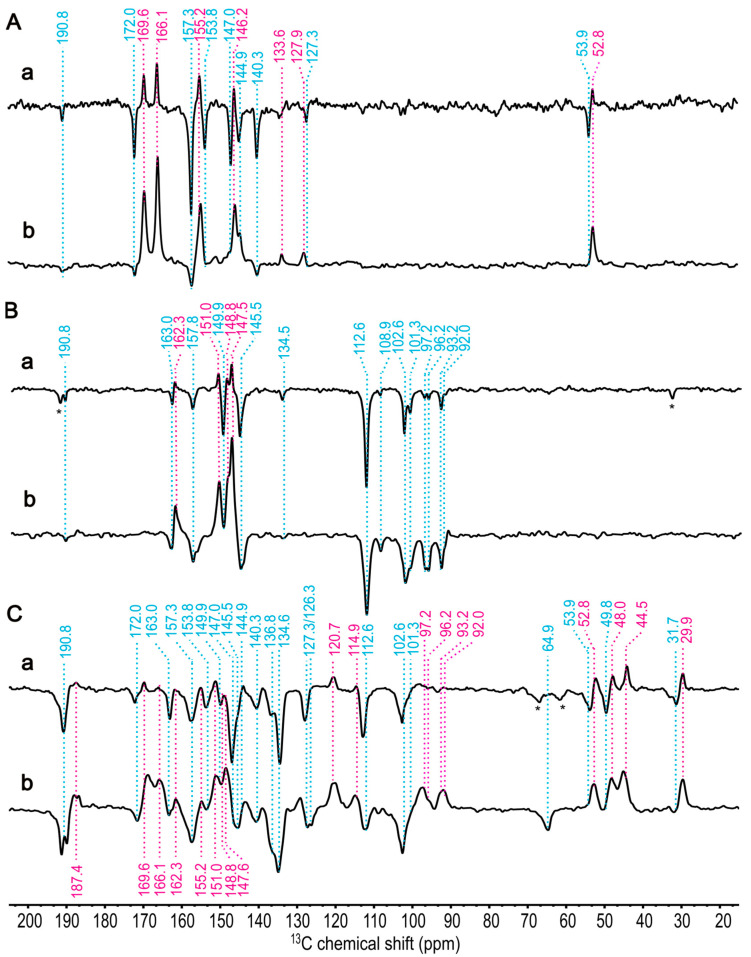
One-dimensional ^13^C photo-CIDNP MAS NMR spectra of [4-^13^C]-ALA obtained at different magnetic fields: (**a**) 9.4 T and (**b**) 4.7 T in (**A**), [5-^13^C]-ALA at (**a**) 9.4 T and (**b**) 4.7 T in (**B**), and [3,4,5-^13^C]-ALA at (**a**) 9.4 T and (**b**) 4.7 T in (**C**) of membrane fragments from *Hb. mobilis* recorded under continuous illumination with 488 nm for 1h. A cycle delay of 4 s at 9.4 T and 2 s at 4.7 T was used, and a MAS frequency of 8 kHz was employed for the experiments conducted at 235 K. Positive signals are indicated in violet; negative signals are marked in blue. The signals labelled with an asterisk are assigned to spinning side band.

**Figure 7 molecules-29-01021-f007:**
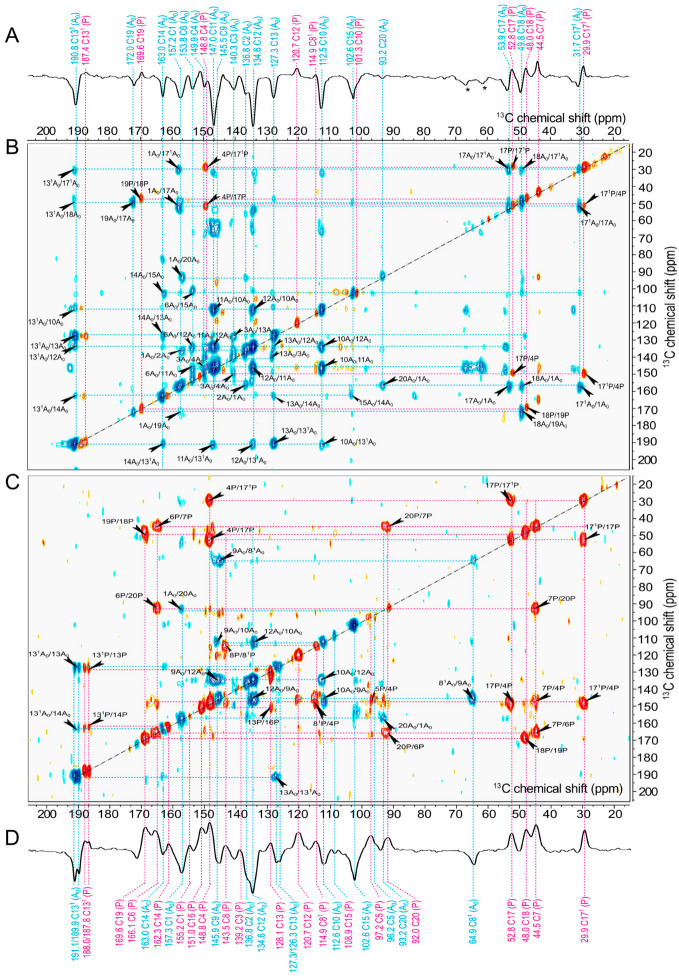
One- and two-dimensional ^13^C-^13^C photo-CIDNP DARR MAS NMR spectra of [3,4,5-^13^C]-ALA-labelled membrane fragments obtained with a spin diffusion mixing time of 50 ms at a MAS frequency of 8 kHz and a temperature of 235 K under the continuous illumination at 488 nm measured at magnetic field strengths of (**A**,**B**) 9.4 T and (**C**,**D**) 4.7 T. Positive contours of the spectra are indicated in red, while negative contours are marked in blue. The signals labelled with an asterisk are assigned to spinning side band.

**Figure 8 molecules-29-01021-f008:**
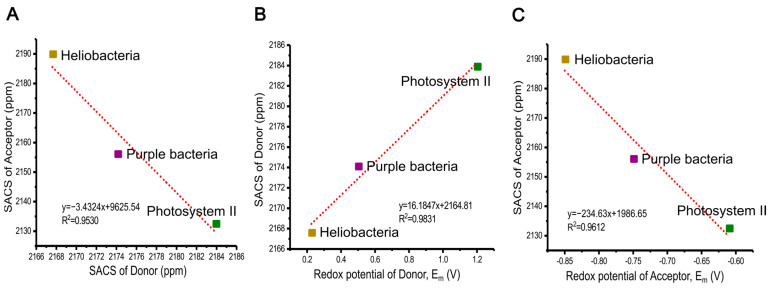
Correlation between the sum of the aromatic chemical shifts (SACS) of the primary donor and acceptor (**A**), between SACS of the primary donor and its redox mid-potential (**B**), and between SACS of primary acceptor and its redox mid-potential, E_m_ (V) (**C**) among the RCs from three different photosynthetic organisms.

**Figure 9 molecules-29-01021-f009:**
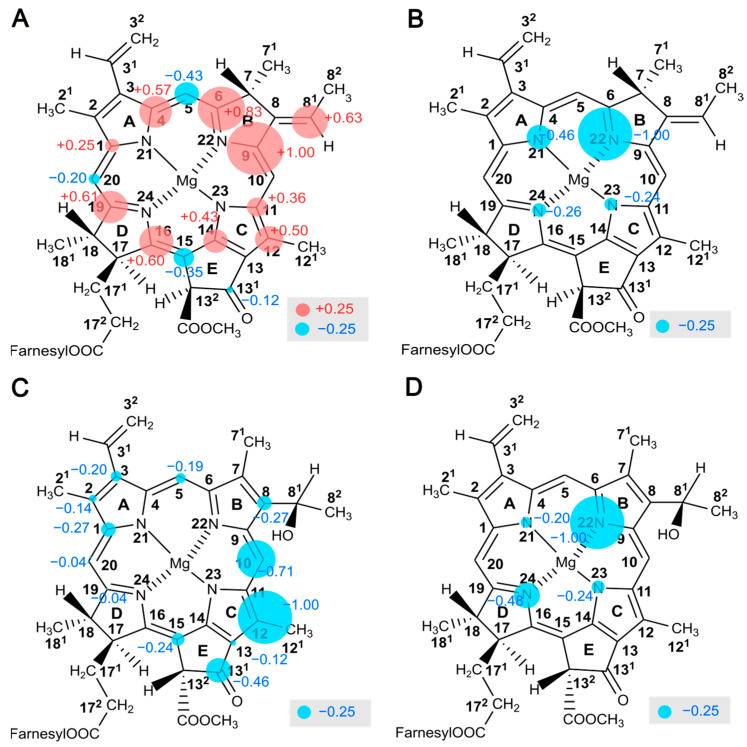
Mapping of local electron spin densities as obtained by solid-state photo-CIDNP intensities. The electron spin density pattern observed of the donor cofactors, BChl *g*′, on ^13^C (**A**) and ^15^N (**B**) nuclei refers to positive signal intensities caused by the dominance of the DR over the TSM mechanism and therefore reflects the local electron spin distribution in the donor triplet state. The electron spin density pattern of the electron acceptor cofactors, 8^1^-OH Chl *a_F_*, on ^13^C (**C**) and ^15^N (**D**) nuclei refers to negative signal intensities caused by the TSM mechanism and therefore reflects the local electron spin distribution in the radical anion state. The size of the circles is semi-quantitatively related to the signal intensity. Positive signals are marked in red, while the negative signals are indicated in blue.

**Table 1 molecules-29-01021-t001:** ^13^C and ^15^N chemical shift assignments for signals observed by ^13^C and ^15^N photo-CIDNP MAS NMR from the membrane fragments of *Hb*. *mobilis*. Listed are the experimental chemical shifts (Expt.), the magnetic shielding constants from DFT calculations (Shield.), and the estimated chemical shift from linear regression (Fitted.). No signal has been identified for C3^1^ from experimental data. Details of the linear regression are given in Appendix A. Labels (*A*) indicate absorptive (positive) and (*E*) emissive (negative) signals, respectively, obtained from experimental data (unit: ppm of chemical shift scale).

Assignment	Electron Donor, BChl *g*′			Electron Acceptor, 8^1^-OH-Chl *a_F_*		
	Expt.	Shield.	Fitted.	Expt.	Shield.	Fitted.
N21	189.0 (*E*)	55.3	190.1	189.0 (*E*)	51.0	190.4
N22	259.6 (*E*)	−18.1	261.2	216.1 (*E*)	26.6	216.8
N23	193.2 (*E*)	53.2	192.2	193.2 (*E*)	50.2	191.3
N24	253.2 (*E*)	−8.1	251.5	249.3 (*E*)	−3.3	249.1
C13^1^	187.4 (*A*)	−14.6	188.6	190.8 (*E*)	−14.5	189.8
C19	169.6 (*A*)	9.9	164.9	172.0 (*E*)	3.9	171.6
C6	166.1 (*A*)	11.4	163.4	153.8 (*E*)	28.2	147.8
C14	162.3 (*A*)	15.8	159.1	163.0 (*E*)	12.6	163.1
C1	155.2 (*A*)	24.8	150.4	157.3 (*E*)	21.9	154.0
C16	151.0 (*A*)	23.9	151.2	157.8 (*E*)	12.6	163.1
C4	148.8 (*A*)	28.5	146.9	149.9 (*E*)	32.1	144.0
C9	147.6 (*A*)	27.8	147.5	145.5 (*E*)	33.6	142.5
C11	146.2 (*A*)	27.9	147.4	147.0 (*E*)	28.4	147.6
C8	143.5 (*A*)	27.7	147.6	144.9 (*E*)	30.5	145.5
C3	139.2 (*A*)	37.4	138.3	140.3 (*E*)	34.4	141.7
C2	133.6 (*A*)	40.3	135.4	136.8 (*E*)	40.7	135.5
C12	120.7 (*A*)	46.5	129.5	134.6 (*E*)	34.9	141.2
C13	127.9 (*A*)	43.5	132.3	127.3 (*E*)	43.2	133.0
C3^1^	_	41.5	134.2	_	41.6	134.6
C10	101.3 (*E*)	79.6	97.4	112.6 (*E*)	63.6	113.0
C15	108.9 (*E*)	65.4	111.2	102.6 (*E*)	72.8	104.0
C5	97.2 (*E*)	81.3	95.7	96.2 (*E*)	73.4	103.4
C20	92.0 (*E*)	83.3	93.8	93.2 (*E*)	86.0	91.0
C8^1^	114.9 (*A*)	57.6	118.7	64.9 (*E*)	113.8	63.7
C17	52.8 (*A*)	128.8	49.8	53.9 (*E*)	129.5	48.3
C18	48.0 (*A*)	130.7	47.9	49.8 (*E*)	125.8	51.9
C7	44.5 (*A*)	134.6	44.2	_	42.2	134.0
C17^1^	29.9 (*A*)	152.0	27.4	31.7 (*E*)	148.0	30.1

**Table 2 molecules-29-01021-t002:** Comparison of the sum of the aromatic chemical shifts (SACS) and overall chemical shift difference (OCSD) between the primary donor and acceptor among the RCs from three different photosynthetic organisms.

Photosynthetic RCs	Primary Electron Donor			Primary Electron Acceptor			OCSD
	Cofactor	SACS (ppm)	E_m_ (V)	Cofactor	SACS (ppm)	E_m_ (V)	Δδ (ppm) (Acc.-Don.)
Heliobacteria	BChl *g*′	2167.6	+0.225 ^a^	8^1^-OH-Chl *a_F_*	2189.9	−0.85 ^d^	+22.3
Purple bacteria	BChl *a*	2174.1	+0.5 ^b^	BPhe *a*	2156.1	−0.75 ^e^	−18.0
Photosystem II	Chl *a*	2183.9	+1.2 ^c^	Phe *a*	2132.5	−0.61 ^f^	−51.5

^a^ refs. [16,39], ^b^ refs. [94,95,96], ^c^ refs. [37,38], ^d^ refs. [39,40], ^e^ refs. [97], ^f^ refs. [98,99].

## Data Availability

The data presented in this study are available on request from the authors (Y.K and J.M.).

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
