# Peer review of "Electronic Structures of Radical-Pair-Forming Cofactors in a Heliobacterial Reaction Center"

_molecules, 2024, doi:10.3390/molecules29051021_

Round 1

Reviewer 1 Report

Comments and Suggestions for Authors

This is an impressive photo-CIDNP study of the Heliobacterial reaction center. The authors are experts in this technique and have done extensive work to identify and characterize the signals observed under strongly reducing conditions. The importance of this latter condition is not explicitly discussed. Presumably, the FeS acceptor will be prereduced, so that the only “acceptor” involved is the earlier chl a-like acceptor. This should be made clear in the manuscript. Also, the possible existence of a quinone-like acceptor has been debated in this system. While this seems to have been mostly ruled out by the X-ray structure, it should still be considered and discussed, as some groups still advocate for this. The early microbiological, biochemical and spectroscopic literature on the heliobacteria is not referenced in an evenhanded manner, and even the discovery of the heliobacteria by Gest is not referenced. The ALA labeling method appears without explanation on p. 7. Most readers will appreciate an introduction to what this is and how the variously labeled ALA precursors end up where they do. To this end, Fig. 8 might be moved to earlier in the text to make this point more clearly. As it is, only a real expert will not get lost at this point in the presentation. The proper IUPAC nomenclature for chlorophylls labels the five rings as A through E, not I through V as does the old Fischer nomenclature. This must be changed.

Author Response

REVIEWER 1

This is an impressive photo-CIDNP study of the Heliobacterial reaction center. The authors are experts in this technique and have done extensive work to identify and characterize the signals observed under strongly reducing conditions. The importance of this latter condition is not explicitly discussed.

Thanks to the reviewer to this very encouraging remark.

Presumably, the FeS acceptor will be prereduced, so that the only “acceptor” involved is the earlier chl a-like acceptor. This should be made clear in the manuscript.

This assumption (pre-reduction of the acceptor) is correct: it is now clarified throughout the manuscript.

Also, the possible existence of a quinone-like acceptor has been debated in this system. While this seems to have been mostly ruled out by the X-ray structure, it should still be considered and discussed, as some groups still advocate for this. The early microbiological, biochemical and spectroscopic literature on the heliobacteria is not referenced in an evenhanded manner, and even the discovery of the heliobacteria by Gest is not referenced.

We thank the reviewer for pointing out the possible role of Q-acceptors. During the discussion of the structure, we now mention this discussion. However, for the present study (all receptors are pre-reduced) this hot question is not decisive.

The ALA labeling method appears without explanation on p. 7. Most readers will appreciate an introduction to what this is and how the variously labeled ALA precursors end up where they do. To this end, Fig. 8 might be moved to earlier in the text to make this point more clearly. As it is, only a real expert will not get lost at this point in the presentation.

Thank you. Based on feedback from other reviewers on ALA labelling and pigment structure, we have moved this information to the "Introduction" section and changed the corresponding figure accordingly (see new Figure 3).

The proper IUPAC nomenclature for chlorophylls labels the five rings as A through E, not I through V as does the old Fischer nomenclature. This must be changed.

In fact, there are (at least three) competing nomenclatures (Fischer, IUPAC, SciFinder). Since our carbon numbering corresponds to IUPAC, we have also adapted the ring numbering to the IUPAC nomenclature (A to E).

Previous studies on other photosynthetic systems have generally used the designations "Ring I" to "Ring V" (including the work of McDermott), but we believe that this change will not cause confusion for readers.

Reviewer 2 Report

Comments and Suggestions for Authors

The manuscript described a study of HbRCs using photo-CIDNP. Utilizing the different mechanisms for signal enhancement of electron donor and acceptor, and the magnetic-field dependence, the authors assigned 13C and 15N peaks for HbRCs in 1D and 2D spectra. The assignment was further supported by DFT calculations. I think it is a comprehensive work.

I would like to know why the 2D 13C-13C correlation spectra did not give a lot of sequential peaks at different mixing times (such as 1P/2P, 2P/3P etc.). Some long distance peaks (17P/4P) showed up even at very short mixing (figure S3, 2.5ms). Please add some explanation in the paper.

I have some minor suggestions.

Line 194, “ the ratio of positive signals to emissive signals remains constant,..” gave a little confusion. I was not clear the ratio indicated the intensity ratio or other factors. The following sentences discussed the intensity changes of individual peaks, which would suggest the ratio changed. Another thing is about using “positive” vs “emissive” as a pair. I prefer using “positive” and “negative” as a pair and “absorptive” and “emissive” as a pair in description of the peaks. The author explained the relationship later in the paper. But here and some other places at the beginning, when it was not discussed, there was a little confusion for me.

Line 222, there is an extra “and”.

Line 456, figure 6, C, D do not have x axis.

Line 630, figure 8, A and B should be the same structure, and C and D should be the same too. Right now A and C are the same.

Author Response

REVIEWER 2

The manuscript described a study of HbRCs using photo-CIDNP. Utilizing the different mechanisms for signal enhancement of electron donor and acceptor, and the magnetic-field dependence, the authors assigned 13C and 15N peaks for HbRCs in 1D and 2D spectra. The assignment was further supported by DFT calculations. I think it is a comprehensive work.

Thank you very much. We are glad to read these helpful and encouraging remarks.

I would like to know why the 2D 13C-13C correlation spectra did not give a lot of sequential peaks at different mixing times (such as 1P/2P, 2P/3P etc.). Some long-distance peaks (17P/4P) showed up even at very short mixing (figure S3, 2.5ms). Please add some explanation in the paper.

The intensity of the diagonal peaks induced by the solid-state photo-CIDNP effect could indeed be the main factor influencing the strength of the cross-peaks. This could explain why no cross peak was observed in subsequent correlations, such as 1P/2P, as the signals from these nuclei have a much lower intensity in the 1D spectrum. However, signals of nuclei such as 4P and 17P show the highest intensity, especially at 4.7 T. Consequently, the cross-peaks appear even at a mixing time of 2.5 ms (Figure S3), which facilitates polarisation transfer over long distances. We thank for this hint and added a statement to the manuscript.

I have some minor suggestions.

Line 194, “ the ratio of positive signals to emissive signals remains constant,..” gave a little confusion. I was not clear the ratio indicated the intensity ratio or other factors. The following sentences discussed the intensity changes of individual peaks, which would suggest the ratio changed.

Thanks. It is clarified in the manuscript.

“The ratio of total observed absorptive peak areas to emissive peak areas between these sample preparations remains constant (the ratio from intact cells: 1.43, membrane fragments: 1.41), suggesting that the contribution of the solid-state photo-CIDNP mechanism remains unchanged.”

Another thing is about using “positive” vs “emissive” as a pair. I prefer using “positive” and “negative” as a pair and “absorptive” and “emissive” as a pair in description of the peaks. The author explained the relationship later in the paper. But here and some other places at the beginning, when it was not discussed, there was a little confusion for me.

We agree. It is revised in the manuscript.

Line 222, there is an extra “and”.

It is revised in the manuscript.

Line 456, figure 6, C, D do not have x axis.

Right. The Figure has been modified in the manuscript.

Line 630, figure 8, A and B should be the same structure, and C and D should be the same too. Right now A and C are the same.

The Figure is corrected in the manuscript.

Reviewer 3 Report

Comments and Suggestions for Authors

Kim et al presented a very interesting study of the electron donor and acceptor cofactors of heliobacterial photosynthetic reaction centers (HbRCs). They assigned the 13C and 15N chemical shifts of the electron donor BChl g’, and electron acceptor 81-hydroxy-Chl aF. The absence of chemical shfit doubling confirms that the two branches of HbRC are indeed symmetrical and the electron transfer occurs in parallel, which distinguishes HbRC from most other RC system. This study fill the gap of our current understanding of evolution of photosynthetic reaction centers.

The use of photo-CIDNP significantly facilitate the assignment. It not only enhances the NMR sensitivity, but also distinguishes signal from donor and acceptor by contribution of different hyperpolariation mechnisms. 2D 13C-13C DARR correlation spectra of the [3,4,5-13C]-ALA-labelled sample provides enough resolution to unambiguously assign the electron donor and acceptor.

Overall, this study is reasonably designed, and the experiment result and analysis is clear. I only have a few questions and suggestions:

1. Consider showing Fig8 and Fig9 earlier, and combine them in some ways. It would help the audience understand the structure of the pigments.

2. Show the chemcial shift axis at the bottom of panel C and D.

3. In line 495-503, the peak splitting is attributed to CC J coupling. Is the same splitting in Hz observed in both 4.7T and 9.4T?

4. In section 2.8, the correlation between SACS of different pigments and redox potential is discussed. Do the DFT calculated chemical shifts also exhibit similar correlation?

Comments on the Quality of English Language

Moderate editing of English language is required. Some phrases are unclear, such as "Therefore, research addressing our enormous energy demands for the future tries to understand the function in de-37 tail."

Author Response

REVIEWER 3

Kim et al presented a very interesting study of the electron donor and acceptor cofactors of heliobacterial photosynthetic reaction centers (HbRCs). They assigned the 13C and 15N chemical shifts of the electron donor BChl g’, and electron acceptor 81-hydroxy-Chl aF. The absence of chemical shfit doubling confirms that the two branches of HbRC are indeed symmetrical and the electron transfer occurs in parallel, which distinguishes HbRC from most other RC system. This study fill the gap of our current understanding of evolution of photosynthetic reaction centers.

The use of photo-CIDNP significantly facilitate the assignment. It not only enhances the NMR sensitivity, but also distinguishes signal from donor and acceptor by contribution of different hyperpolariation mechnisms. 2D 13C-13C DARR correlation spectra of the [3,4,5-13C]-ALA-labelled sample provides enough resolution to unambiguously assign the electron donor and acceptor.

Overall, this study is reasonably designed, and the experiment result and analysis is clear. I only have a few questions and suggestions:

We are very glad to read these very positive comments. Thanks!

  1. Consider showing Fig8 and Fig9 earlier, and combine them in some ways. It would help the audience understand the structure of the pigments.

It is revised in the manuscript.

We rearranged Figure 9, which depicts ALA labelling and pigment structure, to now appear as Figure 3 in the introduction part.

  1. Show the chemcial shift axis at the bottom of panel C and D.

We improved the Fig accordingly.

  1. In line 495-503, the peak splitting is attributed to CC J coupling. Is the same splitting in Hz observed in both 4.7T and 9.4T?

At 4.7 T, the J coupling constant of 131P is about 60 Hz. At 9.4 T, a weak splitting is observed matching this observation. A short comment is now added in the manuscript.

  1. In section 2.8, the correlation between SACS of different pigments and redox potential is discussed. Do the DFT calculated chemical shifts also exhibit similar correlation?

Yes, a clear correlation is observed when we apply the DFT-calculated chemical shifts of HbRC from this study. Both the experimental and calculated chemical shifts show strong correlations (Figure S4). However, since we only have the calculated chemical shifts from DFT for HbRC, we have decided not to include them in the manuscript.

When we applied the SACS of the donor and acceptor of HbRC from DFT (Table 1), we obtained the following correlation coefficients:

  • SACS of donor and acceptor: R2= 0.9767
  • SACS of donor and its redox mid potential: R2=0.9418
  • SACS of acceptor and its redox mid potential: R2= 0.9405

Figure: The chemical shift of HbRC is obtained from DFT, while that of R. sphaeroides WT and PSII (spinach) is obtained from experimental data

Hence, we are convinced that the presented correlations are based on realistic experimental and theoretical data.
